# An Exercise Immune Fitness Test to Unravel Disease Mechanisms—A Proof-of-Concept Heart Failure Study

**DOI:** 10.3390/jcm13113200

**Published:** 2024-05-29

**Authors:** Galyna Bondar, Abhinandan Das Mahapatra, Tra-Mi Bao, Irina Silacheva, Adrian Hairapetian, Thomas Vu, Stephanie Su, Ananya Katappagari, Liana Galan, Joshua Chandran, Ruben Adamov, Lorenzo Mancusi, Isabel Lai, Anca Rahman, Tristan Grogan, Jeffrey J. Hsu, Monica Cappelletti, Peipei Ping, David Elashoff, Elaine F. Reed, Mario C. Deng

**Affiliations:** 1David Geffen School of Medicine, University of California Los Angeles Medical Center, Los Angeles, CA 90095, USA; gbondar@mednet.ucla.edu (G.B.); tbao@mednet.ucla.edu (T.-M.B.); isilacheva@mednet.ucla.edu (I.S.); adrianhairapetian@gmail.com (A.H.); thovu@g.ucla.edu (T.V.); stephaniesu15@g.ucla.edu (S.S.); a.katappagari@gmail.com (A.K.); lianagggalan@gmail.com (L.G.); joshuachandran@g.ucla.edu (J.C.); radamov@g.ucla.edu (R.A.); lmancusi@mednet.ucla.edu (L.M.); ilai@mednet.ucla.edu (I.L.); adrahman@mednet.ucla.edu (A.R.); tgrogan@mednet.ucla.edu (T.G.); jjhsu@mednet.ucla.edu (J.J.H.); mcappelletti@mednet.ucla.edu (M.C.); pping@mednet.ucla.edu (P.P.); delashoff@mednet.ucla.edu (D.E.); ereed@mednet.ucla.edu (E.F.R.); 2Strand NGS, Strand Life Sciences Pvt. Ltd., San Francisco, CA 94104, USA; abhinandan.das@strandls.com

**Keywords:** heart failure, immunological fitness, peripheral blood mononuclear cell transcriptome

## Abstract

**Background**: Cardiorespiratory fitness positively correlates with longevity and immune health. Regular exercise may provide health benefits by reducing systemic inflammation. In chronic disease conditions, such as chronic heart failure and chronic fatigue syndrome, mechanistic links have been postulated between inflammation, muscle weakness, frailty, catabolic/anabolic imbalance, and aberrant chronic activation of immunity with monocyte upregulation. We hypothesize that (1) temporal changes in transcriptome profiles of peripheral blood mononuclear cells during strenuous acute bouts of exercise using cardiopulmonary exercise testing are present in adult subjects, (2) these temporal dynamic changes are different between healthy persons and heart failure patients and correlate with clinical exercise-parameters and (3) they portend prognostic information. **Methods**: In total, 16 Heart Failure (HF) patients and 4 healthy volunteers (HV) were included in our proof-of-concept study. All participants underwent upright bicycle cardiopulmonary exercise testing. Blood samples were collected at three time points (TP) (TP1: 30 min before, TP2: peak exercise, TP3: 1 h after peak exercise). We divided 20 participants into 3 clinically relevant groups of cardiorespiratory fitness, defined by peak VO_2_: HV (*n* = 4, VO_2_ ≥ 22 mL/kg/min), mild HF (HF1) (*n* = 7, 14 < VO_2_ < 22 mL/kg/min), and severe HF (HF2) (*n* = 9, VO_2_ ≤ 14 mL/kg/min). **Results**: Based on the statistical analysis with 20–100% restriction, FDR correction (*p*-value 0.05) and 2.0-fold change across the three time points (TP1, TP2, TP3) criteria, we obtained 11 differentially expressed genes (DEG). Out of these 11 genes, the median Gene Expression Profile value decreased from TP1 to TP2 in 10 genes. The only gene that did not follow this pattern was *CCDC181*. By performing 1-way ANOVA, we identified 8/11 genes in each of the two groups (HV versus HF) while 5 of the genes (*TTC34*, *TMEM119*, *C19orf33*, *ID1*, *TKTL2*) overlapped between the two groups. We found 265 genes which are differentially expressed between those who survived and those who died. **Conclusions**: From our proof-of-concept heart failure study, we conclude that gene expression correlates with VO_2_ peak in both healthy individuals and HF patients, potentially by regulating various physiological processes involved in oxygen uptake and utilization during exercise. Multi-omics profiling may help identify novel biomarkers for assessing exercise capacity and prognosis in HF patients, as well as potential targets for therapeutic intervention to improve VO_2_ peak and quality of life. We anticipate that our results will provide a novel metric for classifying immune health.

## 1. Introduction

All living organisms have developed highly conserved and regulated stress response mechanisms. Strenuous activity leads to the acute activation of the hypothalamic–pituitary–adrenal axis including the adrenergic system [1]. Cardiorespiratory fitness positively correlates with longevity and inversely with all-cause mortality [2,3,4,5,6,7,8,9,10]. The World Health Organization and the American Heart Association are working towards the implementation of cardiorespiratory fitness measurements to improve risk classification and optimize disease prevention [11,12,13,14,15,16].

Cardiorespiratory fitness, defined as the ability of the circulatory and respiratory systems to supply oxygen to skeletal muscles during sustained physical activity correlates with immune health. Maintaining cardiorespiratory fitness exerts beneficial immunological effects [17,18]. It has the potential to delay, halt, or even reverse the age-associated decline in immune function [19,20]. Fitness is hypothesized to be linked to leukocyte gene expression in both healthy individuals and Heart Failure (HF) patients by regulating various physiological processes involved in oxygen uptake and utilization during exercise.

Various data suggest that regular exercise may provide health benefits by reducing systemic inflammation [21]. In chronic disease conditions, such as chronic HF and chronic fatigue syndrome, mechanistic links have been postulated between inflammation, muscle weakness, frailty, catabolic/anabolic imbalance, and aberrant chronic activation of immunity with monocyte upregulation [22,23,24,25,26,27,28]. In these conditions, in which the adrenergic system is permanently activated, levels of circulating cell-free mitochondrial DNA are chronically increased and correlate with secondary organ dysfunction and cell injury. This altered pattern is associated with a pro-inflammatory equilibrium [29,30,31,32].

Cardiorespiratory fitness is measured by peak oxygen uptake (VO_2_) in mL per kilogram body weight per minute during exercise by standardized cardiopulmonary exercise testing (CPX) [33]. CPX has become an important clinical tool to evaluate exercise capacity and predict outcomes in patients with HF and other cardiac conditions. It provides an assessment of the integrative exercise responses involving the pulmonary, cardiovascular, and skeletal muscle systems, which are not adequately reflected through the measurement of individual organ system function.

Our study aims to identify the gap in the detailed molecular signals induced by exercise that might benefit health and prevent disease. While the Molecular Transducers of Physical Activity Consortium is studying healthy individuals with no prior disease condition, there is a gap in focusing on subjects with specific diseases such as HF which this study addresses.

Based on our overall hypothesis that leukocyte gene expression in both healthy volunteers (HV) and Heart Failure (HF) is linked to cardiorespiratory fitness, this paper presents a proof-of-concept data analysis to characterize the relationship between exercise physiology as reflected in the capacity for functional exercise testing by CPX and the whole-transcriptome dynamics of mixed populations of peripheral blood mononuclear cells (PBMC) to characterize the phenotype of healthy persons and HF patients. We specifically hypothesize that (1) temporal changes in transcriptome profiles of PBMC during strenuous acute bouts of exercise using CPX are present in adult subjects, (2) these temporal dynamic changes are different between HV and HF patients and correlate with clinical CPX parameters and (3) they portend prognostic information. We believe that our results may be also applicable to various chronic disease conditions, such as other forms of organ failure, including immune system failure such as Post-Acute Sequelae of COVID (PASC), also called long-Covid, ref. [34] and chronic fatigue syndrome.

## 2. Methods and Design

### 2.1. Subject Population

Twenty persons were included in our proof-of-concept study (4 HV and 16 HF patients) who were evaluated for advanced cardiac care options and consented to UCLA IRB 12-000351. All participants underwent CPX. Blood samples were collected at three Time Points (TP) (Figure 1). 

Subject Recruitment Inclusion criteria: On the basis of NIH-screening criteria (https://clinicalcenter.nih.gov/recruit/index.html accessed on 16 October 2019), we defined for the purpose of our proof-of-concept study HV as persons with no known significant health problems, specifically without any known cardiac, respiratory, or metabolic disease, not taking any chronic prescribed medication. An HF subject was defined as a subject with HF referred to our Advanced Heart Failure and Heart Transplant Program for evaluation of advanced cardiac therapies including heart transplantation. 

Multivariate CPX panel All participants underwent CPX testing using a standardized bicycle ergometer Ramp protocol until each study participant achieved their individual maximum oxygen consumption or peak oxygen uptake (peak VO_2_). The multivariate CPX panel is a validated measure of cardiorespiratory fitness. Peak VO_2_ is reached when oxygen consumption remains at a steady state despite an increase in workload, measured as Respiratory Exchange Ratio (RER) > 1.0. 

Clinical data were collected from questionnaires for HV and from medical records of HF on the day of CPX testing. We divided 20 participants into 3 clinically relevant groups of cardiorespiratory fitness, defined by peak VO_2_: HV (*n* = 4, VO_2_ ≥ 22 mL/kg/min), mild HF (HF1) (*n* = 7, 14 < VO_2_ < 22 mL/kg/min), and severe HF (HF2) (*n* = 9, VO_2_ ≤ 14 mL/kg/min).

### 2.2. Blood Processing

Samples were collected at 3 time points (TP): within 30 min before exercise (TP1), within 60 s of peak exercise (TP2), and within 1 h post-exercise (TP3) into 1 CPT (Becton Dickinson, Franklin Lakes, NJ, USA) tube for RNA-seq analyses. 

RNA analyses Total RNA was isolated from PBMC for each blood sample. Purified RNA quality was verified on an Agilent 2100 Bioanalyzer (Agilent Technologies, Palo Alto, CA, USA). RNA concentrations were determined using a NanoDrop ND-1000 spectrophotometer (NanoDrop Technologies, Wilmington, DE, USA). The mRNA library was prepared with Illumina TruSeq RNA kit (Illumina, San Diego, CA, USA). The cDNA libraries were quantified using Qubit and sequenced on Illumina HiSeq 2500 (Illumina, San Diego, CA, USA). Total mRNA was amplified and sequenced on the whole-genome Illumina HiSeq 3000. 

PBMC sample processing and Gene Expression Profile protocol. Eight mLl of blood was drawn into a CPT tube. Peripheral Blood Mononuclear cells (PBMC) from each sample were purified within 2 h of phlebotomy. The collected blood was mixed and centrifuged at room temperature (22 °C) for 20 min at 2000× *g* RCF. Two mLl of plasma was separated without disturbing the cell layer into an Eppendorf tube (Sigma-Aldrich, St. Louis, MO, USA) and stored at −80 °C for future experiments. The cell layer was collected, transferred to 15 mL conical tubes, and re-suspended in cold Phosphate Buffer Saline (PBS) (Sigma-Aldrich, St. Louis, MO, USA) and centrifuged for 20 min at 300× *g* RCF at 4 °C. The supernatant was aspirated and discharged. The cell pellet was re-suspended in cold PBS, transferred into an Eppendorf tube, and centrifuged for 20 min at 300× g RCF at 4 °C. The supernatant was discharged. The pellet was re-suspended in 0.5 mL RNA Protect Cell Reagent (Qiagen, Valencia, CA, USA) and frozen at −80° C.

PBMC transcriptome RNA sequencing. All samples were processed using next-generation RNA sequencing transcriptome analysis at the UCLA Technology Center for Genomics and Bioinformatics. Briefly, the RNA was isolated from the PBMC using RNeasy Mini Kit (Qiagen, Valencia, CA, USA). The quality of the total RNA was assessed using NanoDrop^®^ ND-1000 spectrophotometer (NanoDrop Technologies, Wilmington, DE, USA) and Agilent 2100 Bioanalyzer (Agilent Technologies, Palo Alto, CA, USA) concentration above 50 ng/µL, purity—260/280~2.0, integrity—RIN > 9.0 and average > 9.5. Then, mRNA library was prepared with Illumina TruSeq RNA kit according to the manufacturer’s instructions (Illumina, San Diego, CA, USA). Library construction consists of random fragmentation of the polyA mRNA, followed by cDNA production using random polymers. The cDNA libraries were quantified using Qubit and size distribution was checked on Bioanalyzer 2100 (Agilent Technologies, Palo Alto, CA, USA). The library was sequenced on HiSeq 2500. Clusters were generated to yield approximately 725 K–825 K clusters/mm^2^. Cluster density and quality were determined during the run after the first base addition parameters were assessed. We performed single-end sequencing runs to align the cDNA sequences to the reference genome. Generated FASTQ files were transferred to the Advanced HF Research Data Center where Avadis NGS 1.5 (Agilent, Palo Alto, CA, USA and Strand Scientific, San Francisco, CA, USA) was used to align the raw RNA-Seq FASTQ reads to the reference genome. After RNA extraction, quantification and quality assessment, total mRNA was amplified and sequenced on the whole-genome Illumina HiSeq 2500. Data were then subjected to DeSeq normalization using NGS Strand/Avadis (v2.1 10 October 2014). 

### 2.3. Statistical Analysis

The 60 single-end samples were aligned with Human Genome (hg38 build) and RNA Seq analysis was performed on the Strand NGS software V4.0. Reads were normalized using DESeq normalization method. We performed filter by expression (20th–100th percentile) on the samples to remove the genes that had very low normalized signal values in all the samples and retained 42,610 entities out of 57,923 entities. We performed fold change analysis with fold change cut off 2.0 and analyzed for all three TP based on all against the single condition. We identified 156 genes expressed in the conditions that satisfied the fold change cut-off. One-way ANOVA was performed on these entities with all TP and the 60 samples. Based on our extensive molecular prediction test development experience, we used the corrected *p*-value cut-off of 0.05, fold change cut-off of 2.0 and Benjamini–Hochberg as multiple testing corrections, retaining 11 genes expressed across all TPs. Subsequently, covariate regression analysis was performed using Pearson similarity metric. In addition, we performed ANOVA for all TP and oxygen uptake in HV, mild HF, and severe HF subjects separately. We were able to identify 5 out of these 11 genes expressed in both healthy and HF subjects.

## 3. Results

### 3.1. Differentially Expressed Genes (DEGs) before/at Max/1 h Post Max Exercise in the Combined Cohort

1.1: First, we were interested in the effect of CPX-based strenuous exercise on the temporal dynamics of PBMC-Gene Expression Profile in the combined cohort (*n* = 20). Based on the statistical analysis with 20–100% restriction, FDR-correction (*p*-value 0.05) and 2.0-fold change across the three time points (TP1, TP2, TP3) criteria, we obtained 11 DEGs using the analytical criteria detailed in the Section 2.3. The biological functions of these 11 genes are summarized in Appendix A. 

1.2: With respect to the directionality of gene expression, we observed that the 11 genes, when ranked by their median expression for each individual study participant and at each of the three TP, showed lower median signal values at TP2 and TP3 compared to TP1. Therefore, we were interested in the temporal dynamics of each of the 11 individual gene activity levels averaged over the cohort of all 20 subjects at each TP (Figure 2).

Out of these 11 genes, the median Gene Expression Profile value decreased from TP1 to TP2 in 10 genes. The only gene that did not follow this pattern was CCDC181 (circled in red). From TP2 to TP3, some GEP reversed their activity to TP1 level (Figure 3).

1.3: Next, we were interested in what extent the 11 strenuous-activity-bout-associated genes identified in the overall cohort were differentially expressed between HV (*n* = 4) and HF (*n* = 16) subjects, applying the same statistical criteria (20–100% restriction, FDR-correction (*p*-value 0.05) and 2.0-fold change across the three TPs). By performing 1-way ANOVA, we identified 8/11 genes in each of the two groups (HV versus HF) while 5 of the genes (*TTC34*, *TMEM119*, *C19orf33*, *ID1*, *TKTL2*) overlapped between the two groups as represented in the Venn diagram, suggesting some biological regulation differences in the PBMC pool (Figure 4). According to further clustering analyses with the 11 genes and their expression in all 20 subjects, we concluded that they did not have a close correlation to the clinical phenotypes (HV, HF1 and HF2 cohorts) at TP1 (resting state) and TP3 (Recovery state) but showed some correlation with HV/HF-state at peak exercise (Figure 5). At peak exercise (TP2), there were two main clusters differentiating severe HF from mild HF and HV and other cluster-mild HF and HV groups.

### 3.2. Differential Behavior of Genes in the HV versus Mild HF1 and Severe HF2 Cohorts (Peak VO_2_/Percent Predicted VO_2_)

2.1: Next, we were interested in understanding the correlation between the PBMC-transcriptome and the major clinical determinants of fitness, i.e., the condition of health, mild or severe HF, during acute bouts of strenuous exercise using CPX. The exploratory analysis suggested positive and inverse correlations. For exploratory reasons, we performed cluster entities on all 20 subjects for peak VO_2_ and percent predicted VO_2_ with the entity list that we derived from the previous fold change analysis (1.5/2) (Figure 6) which showed differences in TP2 samples between mild HF and severe HF groups. With the FC cut off 1.5, the samples for TP2 showed differences between mild HF and severe HF groups. 

### 3.3. Correlations between the DEG/Clinical Profiles (Peak VO_2_/Percent Predicted VO_2_) and Survival/Heart Transplant Outcomes

3.1: Next, we were interested in understanding the correlation between genes associated with death in the severe HF group (n-9), analyzed by time-point (TP1, TP2, TP3) and as shared gene set represented by the Venn diagram (Figure 7). We found 265 genes which are differentially expressed between those who survived and those who died. 

3.2: In order to provide additional support for our hypothesis that the currently identified genes and our previously described and UCLA-patented survival-related PBMC-genes are reflective of an underlying biological process, we compared the DEG-list for those three out of nine severe HF patients who died during follow-up (*n* = 265) with a DEG list derived from 29 AdHF-patient who had undergone MCS during 2012–2014 out of whom 11 died within a year (*n* = 105). 

To reinforce our hypothesis regarding the association of identified genes, including our patented survival-related genes with an underlying biological process, we compared DEG lists. These lists were obtained from three out of nine severe HF patients who died during follow-up in the current study (*n* = 265) and 29 advanced HF patients, 11 of whom died within a year after undergoing mechanical circulatory support surgery in a previous study (*n* = 105) (Bondar, 2017 [35]). Out of these sets of DEG, six genes overlapped (*SPOCD1*, *MUC20*, *OLFM1*, *CACNA1A*, *DDX3Y*, *BCORP1*) (Appendix A). The direction of regulation was the same in five out of six DEGs. The overlap of six DEGs suggests that there may be shared underlying immune-mediated processes related to the risk of dying from Advanced HF requiring further study.

## 4. Discussion 

In our proof-of-concept study, we present data to support our hypotheses that (1) strenuous bouts of acute exercise during CPX induce longitudinal dynamic changes in transcriptome profiles of PBMC in adult subjects, (2) these longitudinal dynamic changes are different between HV and HF patients and correlate with clinical CPX-parameters and (3) they portend potential prognostic information.

First, our data show that longitudinal dynamic changes in PBMC-transcriptome profiles during acute bouts of strenuous exercise using CPX are present in adult subjects confirming prior studies conducted in younger people. Prior studies had suggested that brief exercise alters PBMC gene expression in early- and late-pubertal children. The pattern of change involves diverse genetic pathways, consistent with a global danger-type response, perhaps readying PBMCs for a range of physiological functions from inflammation to tissue repair that, to our knowledge, would be useful following a bout of physical activity [36].

Second, our results suggest that *these longitudinal dynamic changes of each of the 11 DEGs may differ between HV and HF patients*. The question regarding biological relevance needs to equally address (A) why some genes overlap and (B) why some genes do not overlap. We hypothesize, for future studies, that the key to discriminating adaptive versus maladaptive mechanisms of immune health may lie in understanding the patterns of dynamic cell free-mitochondrial DNA clearance and its subsequent effect on the transcriptome and proteome. Of our 11 DEGs, *DPYD-AS1*, *TNFRSF12A*, and *CD300LD* were found to play a role in inflammation and in the immune system [37,38,39]. *TNFRSF12A* was also found to be highly inducible and to play a key role in the development of cardiac hypertrophy followed by HF [40]. We postulate that the temporal dynamics of total cell free-DNA, specifically cell free-mitochondrial DNA, may either be adaptive or maladaptive, depending on the body’s ability to produce and eliminate these molecules. Therefore, in future studies underway in our lab, we propose to characterize the temporal dynamics of the immune system under exercise and resting conditions through the integrated analysis of cell free-DNA quantification (genomic), mRNA expression (transcriptomic), PBMC sub-population and cytokine production (proteomic), as well as clinical (phenomics) data from the multivariate CPX panel.

Third, we identified PBMC transcripts that might portend prognostic outcome prediction information. The fact that mortality/survival-related PBMC transcripts and CPX-related PBMC transcripts did only marginally overlap raises the interesting speculative hypothesis that these phenomena are governed by different biological pathways.

Fourth, differences in gene expression related to VO_2_ peak between healthy individuals and those with HF reflect the underlying pathophysiological changes associated with HF, including alterations in energy metabolism, mitochondrial function, oxygen transport, inflammatory responses, neurohormonal signaling, and structural remodeling. The difference in gene expression related to peak oxygen uptake (VO_2_ peak) between healthy individuals and those with HF is complex and multifactorial. Several studies have investigated gene expression profiles in cardiac and skeletal muscle tissues to understand the molecular mechanisms underlying exercise capacity and its alterations in HF.

We anticipate that our results will provide a novel metric for classifying immune health. We will use innovative multi-omics analyses to better understand this complex evolutionary physiology, which integrates data from different biological layers. Multi-omics profiling may help identify novel biomarkers for assessing exercise capacity and prognosis in HF patients, as well as potential targets for therapeutic intervention to improve VO_2_ peak and quality of life.

### Limitations

First, our proof-of-concept study is very small with 4 healthy volunteers and 16 HF subjects. This limitation implies that the presented data should serve as building blocks for subsequent confirmatory studies. Second, the correlation analyses do not allow any inferences about causality relations between gene expression levels and clinical phenotypes. In addition to larger confirmatory clinical-translational studies, we therefore anticipate the necessity for mechanistic studies.

## 5. Conclusions

In summary, gene expression can influence VO_2_ peaks in both healthy individuals and HF patients by regulating various physiological processes involved in oxygen uptake and utilization during exercise. We found 11 differentially expressed genes that can illustrate changes in activity under exercise conditions and that also can characterize the clinical phenotypes at VO_2_ peak. We also found 265 genes associated with mortality, of which 6 overlap with genes that we found in a mortality cohort in a previous study conducted in our lab. As a whole, gene expression can influence VO_2_ peaks in both healthy individuals and HF patients by regulating various physiological processes involved in VO uptake and utilization during exercise. Dysregulation of gene expression in HF can contribute to reduced VO_2_ peak and exercise intolerance, highlighting the importance of understanding the molecular mechanisms underlying these alterations for developing targeted therapeutic strategies. Further research is needed to elucidate the specific gene expression signatures associated with exercise intolerance in HF to identify potential therapeutic targets for improving exercise capacity in these patients and to test the utility of an immune fitness test to predict outcomes in various disease conditions.

## Figures and Tables

**Figure 1 jcm-13-03200-f001:**
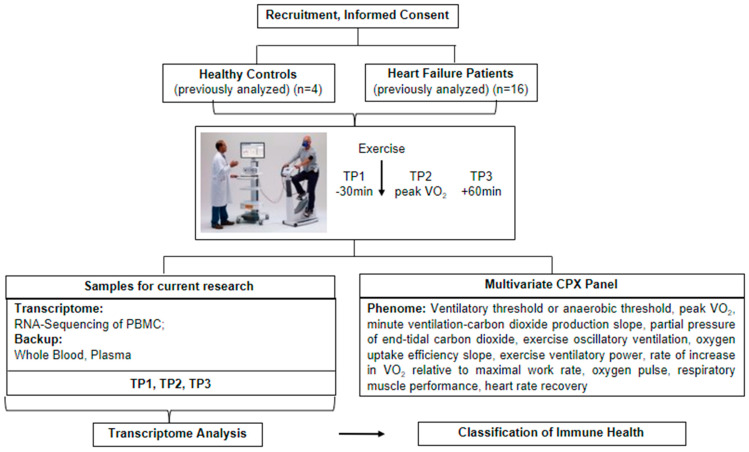
Study Design.

**Figure 2 jcm-13-03200-f002:**
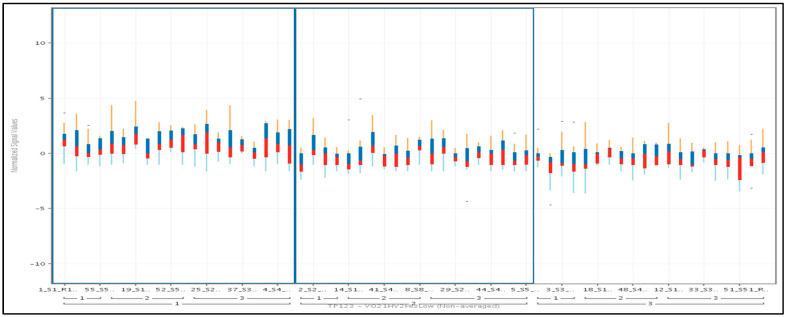
Average expression of 11 genes across all three time points and 20 subjects. Color-coding identifies upper quartile (blue) and lower quartile (red).

**Figure 3 jcm-13-03200-f003:**
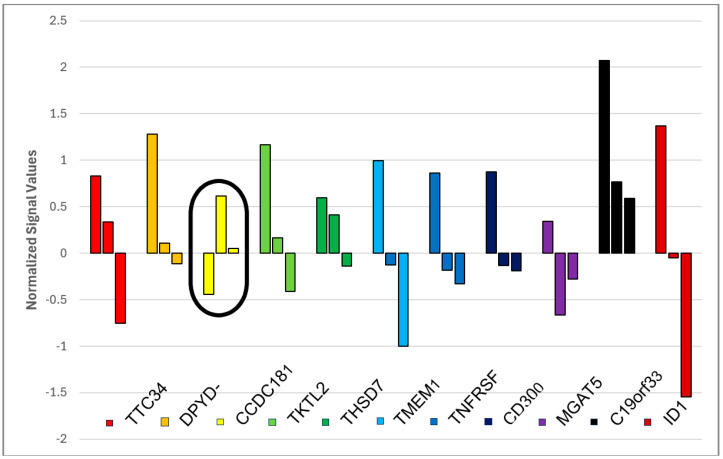
Gene Expression of 11 genes at Time Points 1, 2, and 3. The left columns represent the normalized signal values for Time Point 1, the middle columns for Time Point 2, and the right columns for Time Point 3.

**Figure 4 jcm-13-03200-f004:**
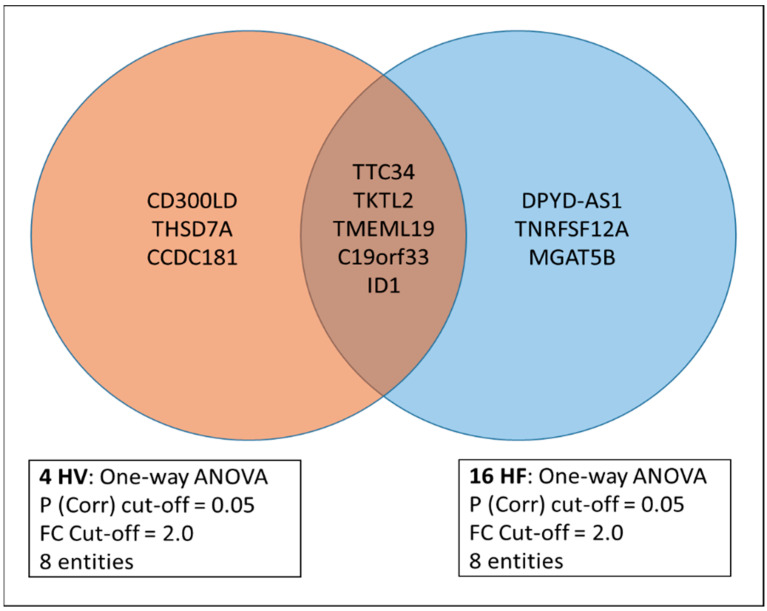
The Venn diagram depicts the 11 genes described in Figure 3 and Appendix A. While 5 of the 11 genes overlap between the group HV and HF, 6 of the 11 genes are specific to either HV (*n* = 4) or HF (*n* = 16).

**Figure 5 jcm-13-03200-f005:**
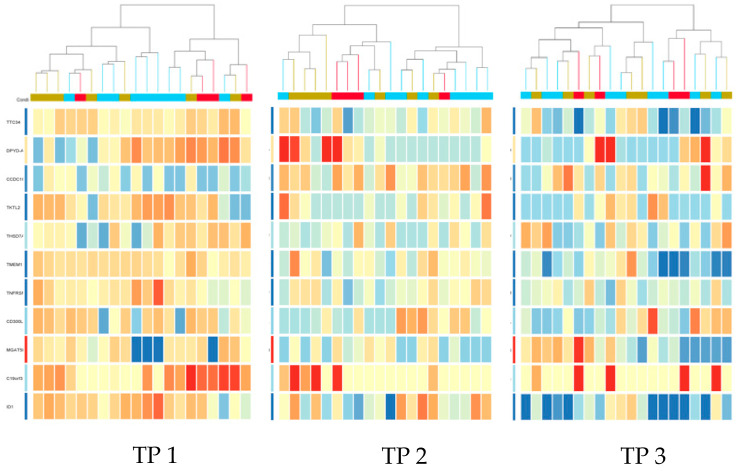
Cluster entities diagram for all 20 subjects and TP1, TP2 and TP3 with FC cut-off 2.0 and 11 genes. Top to bottom: *TTC34*, *DPYD-AS1*, *CCDC181*, *TKTL2*, *THSD7A*, *TMEML19*, *TNRFSF12A*, *CD300LD*, *MGAT5B*, *C19orf33*, *ID1*.

**Figure 6 jcm-13-03200-f006:**
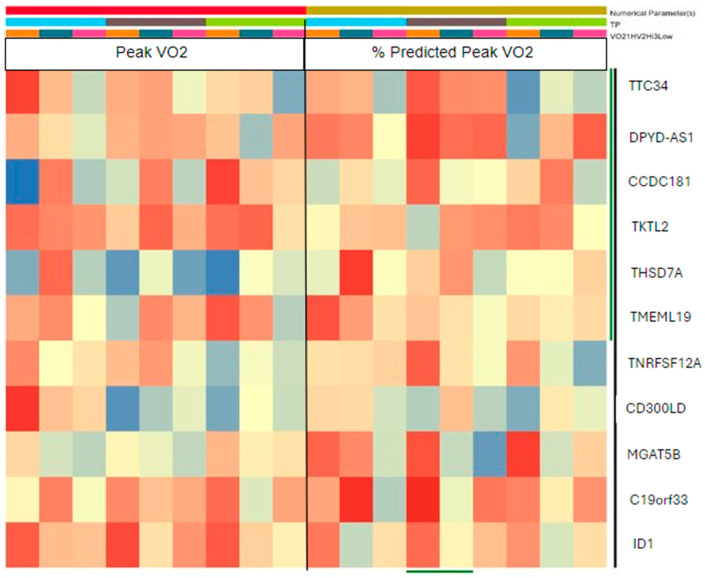
Covariate regression analysis shows an expression of the 11 genes across all three TPs for HV and HF groups in correlation to VO2max and % predicted oxygen uptake. The color coding depicts a positive (red) and a negative (blue) correlation. The darker the color the stronger the respective correlation. Top to bottom: *TTC34*, *DPYD-AS1*, *CCDC181*, *TKTL2*, *THSD7A*, *TMEM119*, *TNFRSF12A*, *CD300LD*, *MGAT5B*, *C19orf33*, *ID1*.

**Figure 7 jcm-13-03200-f007:**
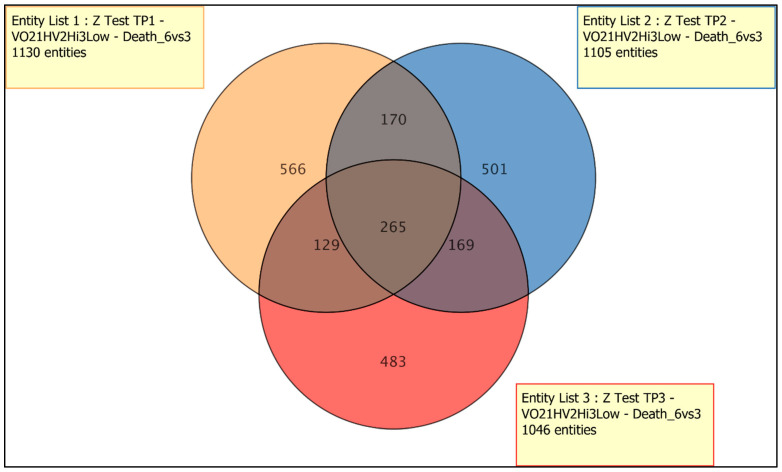
HF group (n-9), analyzed by time-point (TP1, TP2, TP3) and as a shared gene set. Comparison of Z test results for death-associated heart failure group and three time points separately.

## Data Availability

Data is contained within the article and Appendix A.

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
