# Peer review of "An Exercise Immune Fitness Test to Unravel Disease Mechanisms—A Proof-of-Concept Heart Failure Study"

_jcm, 2024, doi:10.3390/jcm13113200_

Round 1

Reviewer 1 Report

Comments and Suggestions for Authors

In the manuscript titled “An Exercise Immune Fitness Test to Unravel Disease Mechanisms—A Proof-Of-Concept Heart Failure Study,” the authors evaluate the PBMC transcriptome of heart failure patients and healthy patients before, during, and after cardiopulmonary exercise. After this, the authors analyze the transcriptomic changes between conditions and between time points.

The authors have an interesting project that could result in valuable findings and currently need some polishing to improve the communication of the results obtained.

Major Comments and Suggestions

- Current conclusion (Line 37 – 40) states that “we conclude that gene expression can influence VO2 peak in both healthy individuals and HF patients by regulating various physiological processes involved in oxygen uptake and utilization during exercise.” This is not supported by the current data as influence would indicate some sort of causative relationship, which has not been explored at this time. This is acknowledged by the authors in the limitations section.

- Making the RNA sequencing data publicly available would be good practice. I would suggest the authors upload it to GEO.

- Table 1 should be added as supplementary information. Currently, it doesn’t provide sufficient value to be in the main text.

- Gene nomenclature for human genes is all capitalized and italics, e.g. SPOCD1.

- Table 2 should be added as supplementary information.

Minor Comments and Suggestions

Line 95 – Figure 1 has no in-text figure legend. You should remove the “Figure 1: Study design” from the figure and add it to the text. Text in the figure (“Transcriptome Analysis” and “Classification of Immune Health”) is distorted.

Line 189 – Figure 2 has no in-text figure legend. Text should be removed from the figure image and added to the text. Y-axis font should be edited to remove distortion.

Line 191 – Figure 3 has no in-text figure legend. Text should be removed from the figure image and added to the text. X-axis labels should be underneath the graph.

Line 206 – Figure 4 has no in-text figure legend—same comment as above.

Line 208 – Figure 5 has no in-text figure legend—same comment as above. Genes on the Y-axis are distorted and not legible.

Line 223 – Figure 6 has no in-text figure legend—same comment as above. Current figure legend appears to be cut.

Line 231 – Figure 7 has no in-text figure legend—same comment as above. Current figure legend appears to be cut.

Comments on the Quality of English Language

N/A

Author Response

Reviewer 1

Summary: In the manuscript titled “An Exercise Immune Fitness Test to Unravel Disease Mechanisms—A Proof-Of-Concept Heart Failure Study,” the authors evaluate the PBMC transcriptome of heart failure patients and healthy patients before, during, and after cardiopulmonary exercise. After this, the authors analyze the transcriptomic changes between conditions and between time points. The authors have an interesting project that could result in valuable findings and currently need some polishing to improve the communication of the results obtained.

Major Comments and Suggestions

- Current conclusion (Line 37 – 40) states that “we conclude that gene expression can influence VO2 peak in both healthy individuals and HF patients by regulating various physiological processes involved in oxygen uptake and utilization during exercise.” This is not supported by the current data as influence would indicate some sort of causative relationship, which has not been explored at this time. This is acknowledged by the authors in the limitations section. We agree with Reviewer 1 and have modified the text as follows: “Conclusions: From our proof-of-concept heart failure study, we conclude that gene expression correlates with VO2 peak in both healthy individuals and HF patients, potentially by regulating various physiological processes involved in oxygen uptake and utilization during exercise.”

- Making the RNA sequencing data publicly available would be good practice. I would suggest the authors upload it to GEO. We agree with Reviewer 1 and have uploaded our raw data to GEO.

- Table 1 should be added as supplementary information. Currently, it doesn’t provide sufficient value to be in the main text. We agree with Reviewer 1 and have moved Table 1 to Supplementary Information.

- Gene nomenclature for human genes is all capitalized and italics, e.g. SPOCD1. We agree with Reviewer 1 and have capitalized/italized gene names.

- Table 2 should be added as supplementary information. We agree with Reviewer 1 and have moved Table 2 to Supplementary Information.

Minor Comments and Suggestions

Line 95 – Figure 1 has no in-text figure legend. You should remove the “Figure 1: Study design” from the figure and add it to the text. Text in the figure (“Transcriptome Analysis” and “Classification of Immune Health”) is distorted. We agree with Reviewer 1 and have moved the Figure 1 text to in-text legend.

Line 189 – Figure 2 has no in-text figure legend. Text should be removed from the figure image and added to the text. Y-axis font should be edited to remove distortion. We agree with Reviewer 1 and have moved the Figure 2 text to in-text legend.

Line 191 – Figure 3 has no in-text figure legend. Text should be removed from the figure image and added to the text. X-axis labels should be underneath the graph. We agree with Reviewer 1 and have moved the Figure 3 text to in-text legend.

Line 206 – Figure 4 has no in-text figure legend— We agree with Reviewer 1 and have moved the Figure 4 text to in-text legend.

Line 208 – Figure 5 has no in-text figure legend— We agree with Reviewer 1 and have moved the Figure 5 text to in-text legend.

Line 223 – Figure 6 has no in-text figure legend— We agree with Reviewer 1 and have moved the Figure 6 text to in-text legend.

Line 231 – Figure 7 has no in-text figure legend— We agree with Reviewer 1 and have moved the Figure 7 text to in-text legend.

Reviewer 2 Report

Comments and Suggestions for Authors

Summary

The manuscript describes a proof-of-concept study examining how the transcriptome profiles of peripheral blood mononuclear cells (PBMCs) change over time during short bouts of exercise using cardiopulmonary exercise testing (CPX) in healthy volunteers and people with heart failure. The study aims to correlate these changes with clinical exercise parameters and explore their predictive potential.

Introduction and objectives

  • Relevant literature supports the introduction's solid theoretical framework, which links exercise to immune system modulation and cardiorespiratory fitness.
  • Question:  The introduction could be improved by providing more specific information about the gaps this study aims to fill compared to previous studies. A clearer delineation of the study's unique contributions to the field would help strengthen the rationale.

Methods

  • The detailed description of the exercise protocol, blood sample collection, and RNA sequencing is a strong point, offering transparency and replicability. Using a multivariate CPX panel for assessing cardiorespiratory fitness is well justified.
  • Request: The statistical methods section lacks some specifics about the analytic techniques and justifications for the chosen methods, such as the reasoning behind selecting a 2.0-fold change as a cutoff. Clarifying these choices could enhance the methodological rigor.

Results

 The results are clearly structured, logically presenting findings at different time points and among different participant groups. Identifying 11 differentially expressed genes and their potential implications are interesting and relevant.

Question: The results would benefit from more detailed statistical analysis, including confidence intervals and effect sizes for the changes observed. Furthermore, the absence of figures in the provided text restricts the comprehensive evaluation of the data presentation.

Discussion

The discussion effectively connects the study findings with broader implications for heart failure management and potential therapeutic interventions. It is forward-thinking to consider multi-omics approaches to enhance understanding of exercise-induced immune changes.

Request:   The discussion could further explore the limitations of the study, such as the small sample size and its impact on the generalizability of the findings. More in-depth speculation on the mechanisms behind the observed gene expression changes could enrich the narrative.

Conclusion

  • The conclusion briefly summarizes the findings and underscores the potential of exercise-induced gene expression profiles as biomarkers for cardiovascular fitness and prognosis in heart failure.
  • Appointment: The conclusions could offer more direct implications for future research directions and clinical practice, particularly regarding how these findings might influence specific interventions or therapeutic approaches.

Author Response

Reviewer 2

Summary

The manuscript describes a proof-of-concept study examining how the transcriptome profiles of peripheral blood mononuclear cells (PBMCs) change over time during short bouts of exercise using cardiopulmonary exercise testing (CPX) in healthy volunteers and people with heart failure. The study aims to correlate these changes with clinical exercise parameters and explore their predictive potential.

Introduction and objectives

  • Relevant literature supports the introduction’s solid theoretical framework, which links exercise to immune system modulation and cardiorespiratory fitness.

Question:  The introduction could be improved by providing more specific information about the gaps this study aims to fill compared to previous studies. A clearer delineation of the study’s unique contributions to the field would help strengthen the rationale. We agree with Reviewer 2 and have modified the text accordingly, including the sentence: ”Our study is targeting to identify the gap in the detailed molecular signals induced by exercise that might benefit health and prevent disease. While the Molecular Transducers of Physical Activity Consortium is studying healthy individuals with no prior disease condition, there is a gap in focusing on subjects with specific diseases such as HF which this study addresses. Based on our overall hypothesis that leukocyte gene expression in both healthy volunteers (HV) and Heart Failure (HF) is linked to cardiorespiratory fitness, this paper presents a proof-of-concept data analysis to characterize the relationship between exercise physiology as reflected in the capacity for functional exercise testing by CPX and the whole-transcriptome dynamics of mixed populations of peripheral blood mononuclear cells (PBMC) to characterize the phenotype of healthy persons and HF patients.”

Methods

  • The detailed description of the exercise protocol, blood sample collection, and RNA sequencing is a strong point, offering transparency and replicability. Using a multivariate CPX panel for assessing cardiorespiratory fitness is well justified.
  • Request: The statistical methods section lacks some specifics about the analytic techniques and justifications for the chosen methods, such as the reasoning behind selecting a 2.0-fold change as a cutoff. Clarifying these choices could enhance the methodological rigor. We agree with Reviewer 2 and have modified the text accordingly, including the sentence: “Based on our extensive molecular prediction test development experience, we used the corrected p-value cut-off of 0.05, fold change cut-off of 2.0 and Benjamini-Hochberg as multiple testing correction”.

Results

 The results are clearly structured, logically presenting findings at different time points and among different participant groups. Identifying 11 differentially expressed genes and their potential implications are interesting and relevant.

Question: The results would benefit from more detailed statistical analysis, including confidence intervals and effect sizes for the changes observed. Furthermore, the absence of figures in the provided text restricts the comprehensive evaluation of the data presentation. We agree with Reviewer 2 and have added the Figures in the text.

Discussion

The discussion effectively connects the study findings with broader implications for heart failure management and potential therapeutic interventions. It is forward-thinking to consider multi-omics approaches to enhance understanding of exercise-induced immune changes.

Request:   The discussion could further explore the limitations of the study, such as the small sample size and its impact on the generalizability of the findings. More in-depth speculation on the mechanisms behind the observed gene expression changes could enrich the narrative. We agree with Reviewer 2 and have modified the Limitations Section as follows: “First, our proof-of-concept study is very small with 4 healthy volunteers and 16 HF subjects. This limitation implies that the presented data should serve as building blocks for subsequent confirmatory studies. Second, the correlation analyses do not allow any inferences about causality relations between gene expression levels and clinical phenotypes. In addition to larger confirmatory clinical-translational studies, we therefore anticipate the necessity for mechanistic studies.

Conclusion:  The conclusion briefly summarizes the findings and underscores the potential of exercise-induced gene expression profiles as biomarkers for cardiovascular fitness and prognosis in heart failure.

  • Appointment: The conclusions could offer more direct implications for future research directions and clinical practice, particularly regarding how these findings might influence specific interventions or therapeutic approaches. We agree with Reviewer 2 and have modified the Conclusion Section accordingly, including the sentence: “Further research is needed to elucidate the specific gene expression signatures associated with exercise intolerance in HF, to identify potential therapeutic targets for improving exercise capacity in these patients and to test the utility of an immune fitness test to predict outcomes in various disease conditions.”.

Reviewer 3 Report

Comments and Suggestions for Authors

Research is interesting and innovative!

However, a very small sample of patients and a slightly smaller number of healthy controls! why on such a small one the number of sick and healthy controls is not at least approximately equal to the sample 16:16!

It is necessary to expand the study or limit it for a series of cases!

The results are nicely presented and the conclusion is drawn from the research!

Reduce the number of references, especially in the introduction!

It is necessary to expand the inclusion and exclusion criteria and state the lack of the study!

Author Response

Reviewer 3

Summary: Research is interesting and innovative!

However, a very small sample of patients and a slightly smaller number of healthy controls! why on such a small one the number of sick and healthy controls is not at least approximately equal to the sample 16:16! It is necessary to expand the study or limit it for a series of cases! We agree with Reviewer 3 and have plans to move beyond the current proof-of-concept study by increasing the samples size of HV and HF subjects.

The results are nicely presented and the conclusion is drawn from the research!

Reduce the number of references, especially in the introduction! We feel that the references in the Introduction are meaningfully explaining the study rationale.

It is necessary to expand the inclusion and exclusion criteria and state the lack of the study! We agree with Reviewer 3 and have expanded the text including the sentence: “A HF subject was defined as a subject with HF referred to our Advanced Heart Failure & Heart Transplant Program for evaluation of advanced cardiac therapies including heart transplantation. ”.

Round 2

Reviewer 3 Report

Comments and Suggestions for Authors

No